# Modeling of pump performance in a water pumping plant
**Fouad LAAJINE[1,2], Mohammed MACHKOR[2], Driss MAZOUZI[1]**
[1]LRNE, multidisciplinary faculty of Taza, Sidi Mohamed Ben Abdellah University,
Fez, Morocco
[2]National Office of Electricity and Drinking Water Morocco
Correspondence: Driss MAZOUZI (driss.mazouzi@usmba.ac.ma)
**ABSTRACT:**
Energy use in drinking Water Supply System represents an important part of the
global energy consumption across all sectors. This portion is expected to raise, due to the
raising demand and the recourse to unconventional water resources. For the water utilities,
most of their operating costs are related to energy consumptions, especially the consumption
of pumping systems. The main objective of this study is to produce a model which reflects
the real behaviour of a pumping system to help in taking decisions on which pump to use
First and which one to replace in case of a limited renovation. In order to do so, Multiple
Linear regression was fitted to model the ratio kWh/m$^3$ produced depending on the input
parameters. The final model describes in a good manner the phenomenon ($R^2$=0.91), so it can
be a good estimator as the calculated ratio is close to the experimental one. The Novelty of
this approach is to have a model which takes into account the real behaviour of the system
whereas most of the studies focus on the pump scheduling problem.

**Key Words:** Energy Efficiency, Linear Multiple regression, Pumping Systems, Water
Supply.
**INTRODUCTION:**
Water and energy are essential ingredients of life. Without them, life would not be
possible. In the upcoming years and decades, water demand is forecasted to increase at a
significant rate of 1% to reach 35% in the year 2050 (UN-Water, 2020) as compared to now
and the worldwide energy consumption is expected to increase by 30% (EIA, 2019). Several
studies have concluded that the energy that is consumed in the pumping process accounts for
7% of the total energy used across the globe (Coelho, 2014). This share is expected to get
bigger due to the increasing distances between the resources and the populations, especially
in water-scarce countries and the growing consumption per capita due to the improvements
of the standards of living and industrialization.
With the current 2030 Agenda to generalize the access to drinking water supply as
part of the Sustainable Development Goal 6 (SDG6), it is very important to keep the tariffs
of the water affordable to the population. To keep the prices of water low, the utilities should
reduce the production costs and hence reduce the energy consumption which is typically the
largest marginal costs for the production (Helena Mala-Jetmarova, 2017).
Pumps account for 80% to 90% of the energy consumption (Sarbu, 2016). By achieving
energy efficiency improvements measures, we can reduce the consumption by at least 25%
(Moreira, 2013). Very few studies were conducted before to simulate the real behaviour of
pumping systems and evaluate the influence of parameters such as the aging of the
components, which can induce a reduction of the pumps performance for up to 12% (Kaya,
45 2008).

In this perspective, this work targets the pumping system of the Bab Louta drinking
water production in the province of Taza, Morocco. A multiple linear regression was
conducted to determine the influence of the parameters on the ratio of kWh consumed per
cubic meter produced.
**1. MATERIALS AND METHODS**
Conventional water supply systems (WSS) consist of sets of structures and facilities
to provide a product with the quantity and the quality that is suitable for domestic and
industrial use. A WSS must be evaluated in terms of mass and energy to develop an energy
and hydraulic model as shown in figure 1 (Vilanova, 2014).

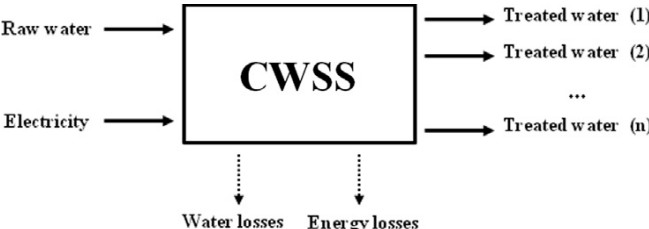


**Figure 1:** Energy and Hydraulic flows in a WSS
There are various methods to enhance energy efficiency in WSS, ranging from
simple monitoring operations of controlling leakages to massive investments consisting of
reviewing the design of the infrastructures or the upgrade of the equipment, to more efficient
ones, passing by pump system optimization and real time control. These Methods were
classified into 3 major sub-categories in figure 2 (Kalaiselvan, 2016).

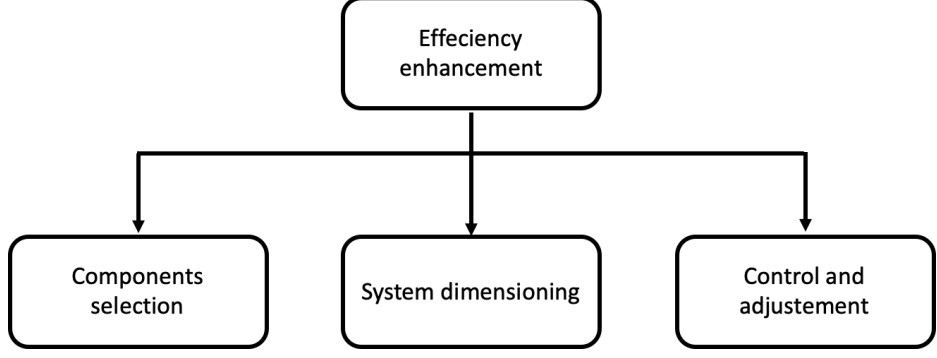


**Figure 2:** Efficiency enhancement opportunities in WSS
**1.1. Area of the study and description of the drinking water production system:**
The province of Taza is located in the centre North of Morocco and it is one of the 9
provinces of the region of Fez-Meknes with a population of roughly 530 000 inhabitants
(Morocco, 2014). The water treatment plant of Tahla provides a large population of the
province, mainly the urban areas. It is situated 60 km from the city of Taza. The plant treats
the raw water of the Bab Louta reservoir.
The production system consists of a pumping station SP0 for raw water, a water
treatment plant then another pumping station of SP3 to reach the city of Taza, which is the
chief town of the province.

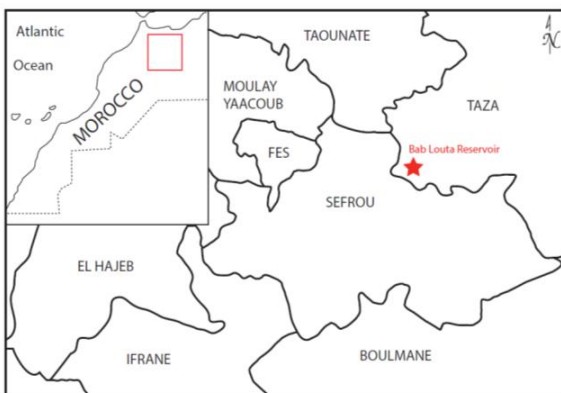


**Figure 3:** Location of the Bab Louta reservoir

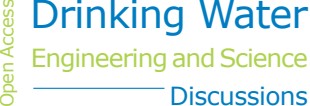

**1.2. Description of the pumping station (object of the study):**
The pumping station SP0 is located about 3km from the reservoir. It is responsible for
overcoming the difference in altitude between the raw water intake and the water treatment
plant. The water is taken from the reservoir to the pumping station SP0 by gravitation
through a pipe of Nominal Diameter of 700mm then pumped through a pipe of a Nominal
Diameter of 600 mm to a tank RMC0 with a capacity of 1000 m$^3$ which provides the water
treatment plant with raw water through a pipe of Nominal Diameter of 500mm (ONEE,

82    2019).

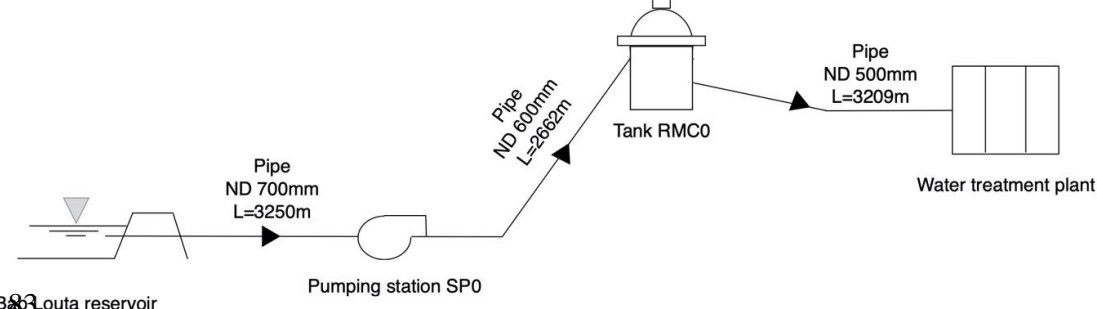

**Figure 4:** Location of the Bab Louta reservoir
The pumping station which is studied here in this article is situated at an altitude of
498.20 m and RMC0 is situated at an altitude of 737.80 m SP0 consists of four (4) pumps
with a 457 m$^3$/h and a head of 209 m. They are manufactured by the company
HIDROTECAR powered by an electric motor which is manufactured by LEROY SOMER
with a nominal power of 455 kW (ONEE, 2019).

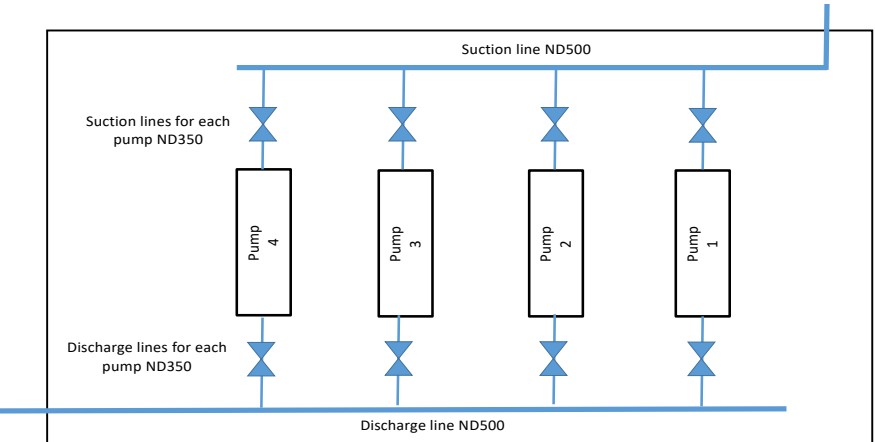

**Figure 5:** Hydromechanical scheme of the pumping station SP0

### 1.3. Experimental procedure:

To obtain a reliable model of our system, measurements of the key parameters influencing it, were conducted. The measurements were taken in the period between January 2015 and December 2018 on a daily basis for the parameters:

- $E_P$: the active energy consumed by the pumping station measured by a wattmeter;

- $E_Q$: the reactive energy consumed by the pumping station measured by a wattmeter;

- V: the volume produced each day measured by an electromagnetic flowmeter;

- Cosφ: the power factor measured;

- HMGi: the operating time of the pump "i" measured by a clock.

### 1.4. Modelling:

The aim of this study is to use Multiple linear regression, a wide popular technique to predict an output from a range of inputs. MLP model with multiple input variables can be expressed as following (Longo, 2016):

$$Y=\beta_0+\beta_1X_1+\beta_2X_2+.....+\beta_nX_n \qquad\qquad Eq\ 1$$
where:
- Y is the input Variable;
- $\beta_i$ are the regression parameters;
- $X_i$ are the input variables.

In order to assess the influence of the parameters on the ratio of the cubic meters produced by the pumping station , the parameters given below were considered: the Active energy (EP), the reactive energy (EQ), the volume produced each day (V), the power factor (Cosφ), the operating time of each pump (HMGi).

**Table 1:** Problem characteristics

| Objective of the study | The effects |
|---|---|
| Number of Variables | 8 |
| Number of experiments | 1388 |
| Number of the coefficients | 8 |
| Number of responses | 1 |

The table above summaries the objective of the study evaluating the effects of 8 variables on the response, which is the ratio of Kwh/$m^3$ produced. To get enough data, 1388 experiments were conducted during a period of 4 years.

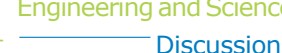
122                     **Table 2:** Measurements Summary

| | $E_P$ (kWh) | $E_Q$(kVar) | Cos Phi | Production ($m^3$) | HMG1 (m) | HMG2 (m) | HMG3 (m) | HMG4 (m) | Ratio (kWh/$m^3$) |
|---|---|---|---|---|---|---|---|---|---|
| Mean | 14 837.47 | 8 339.58 | 0.87 | 18 181.40 | 7.95 | 7.63 | 10.43 | 9.01 | 0.82 |
| Standard deviation | 9 158.63 | 8 529.85 | 0.05 | 5 081.13 | 8.90 | 5.88 | 11.51 | 8.85 | 0.43 |

123         In the Table above there is a preview of the mean and the standard deviation of the

124         parameters measured.

125         In the graphs below the distribution of the parameters is represented.

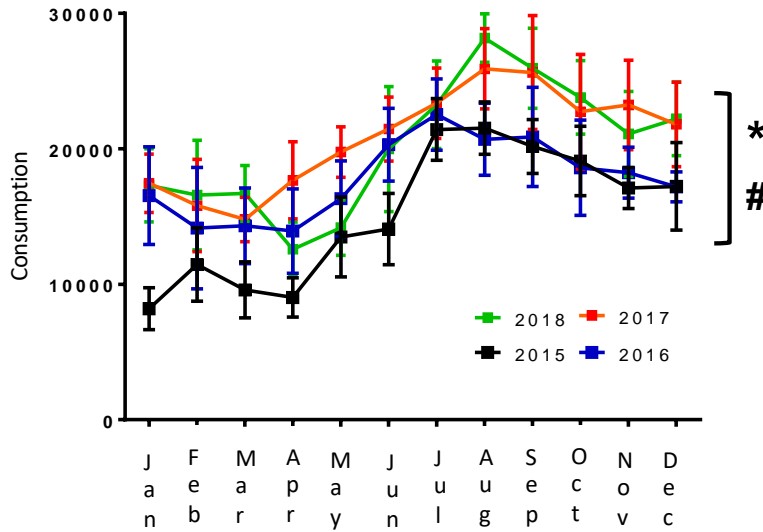


127         **Figure 6:** Representation of the variation of the consumptions through the years


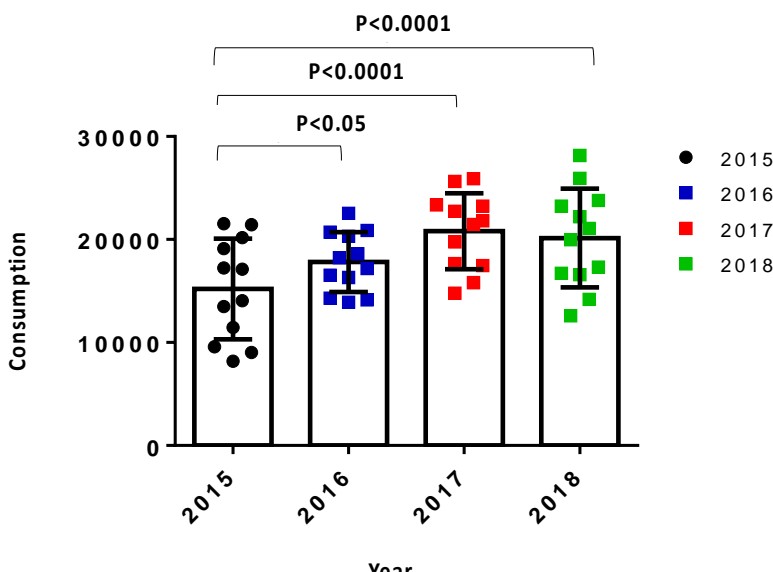

**Figure 7:** Box plots of the consumptions through the years
**2. RESULTS AND DISCUSSION:**
**2.1. Statistical interpretation:**
From the correlation matrix (Table 3), it is found that the variables are strongly
correlated in fact:
• P is highly dependent upon the production First, then the reactive energy and
after come the operating hours of the pumps 4,1 then the cos phi and at the end
the HMG2 and HMG3 respectively.
• Q is highly dependent upon the production First, then the active energy and
after comes the cos phi then at the end the operating hours of the pumps 4,1,2,3
respectively.
• Cosphi is highly dependent upon the reactive energy First then the production,
right after there's the active energy and after come the operating hours of the
pumps 4,1,3,2 respectively.
• Prod is highly dependent upon the active energy P then the reactive energy Q
then after come the operating hours of the pumps 4,1,2,3.





**Table 3:** Correlation Matrice

| Variable | Means | Std. Dev. | Correlations (MATRICE CR without outliers) Marked correlations are significant at p < 0,05000 N=1388 (Casewise deletion of missing data) | | | | | | | | |
| | | | P | Q | Cos phi | Prod | HMG1 | HMG2 | HMG3 | HMG4 | Ratio |
|---|---|---|---|---|---|---|---|---|---|---|---|
| **P** | -0.022423 | 0.487229 | 1.000000 | 0.883071 | -0.190305 | 0.939746 | 0.333340 | 0.202424 | 0.169873 | 0.625872 | 0.463030 |
| **Q** | -0.027636 | 0.345041 | 0.883071 | 1.000000 | -0.585775 | 0.883506 | 0.332525 | 0.170169 | 0.170505 | 0.612964 | 0.298339 |
| **Cos phi** | 0.026763 | 0.761882 | -0.190305 | -0.585775 | 1.000000 | -0.316528 | -0.197066 | 0.024943 | -0.076562 | -0.271961 | 0.193819 |
| **prod** | 0.000741 | 0.973866 | 0.939746 | 0.883506 | -0.316528 | 1.000000 | 0.340309 | 0.222533 | 0.175107 | 0.613597 | 0.150594 |
| **HMG1** | -0.001400 | 1.002171 | 0.333340 | 0.332525 | -0.197066 | 0.340309 | 1.000000 | -0.164919 | -0.016355 | 0.236341 | 0.111733 |
| **HMG2** | -0.001865 | 0.998758 | 0.202424 | 0.170169 | 0.024943 | 0.222533 | -0.164919 | 1.000000 | -0.086627 | -0.141591 | 0.006575 |
| **HMG3** | 0.001362 | 1.003257 | 0.169873 | 0.170505 | -0.076562 | 0.175107 | -0.016355 | -0.086627 | 1.000000 | 0.074751 | 0.036371 |
| **HMG4** | -0.017745 | 0.708400 | 0.625872 | 0.612964 | -0.271961 | 0.613597 | 0.236341 | -0.141591 | 0.074751 | 1.000000 | 0.0257315 |
| **Ratio** | -0.051987 | 0.197427 | 0.463030 | 0.298339 | 0.193819 | 0.150594 | 0.111733 | 0.006575 | 0.036371 | 0.0257315 | 1.000000 |



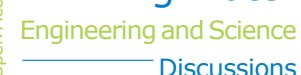

From the table of the regression summary (Table 4) it is conclude that the factors
influencing the ratio in a descending order are:
• Ratio is positively correlated with the active energy consumed by the pumps;
• Ratio is negatively correlated with the production;
• Ratio is positively correlated with the CosPhi;
• Ratio is negatively correlated with the reactive energy consumed by the
pumps;
• Ratio is positively correlated with the operating hours of the pumps 1 and 4.
**Table 4:** Regression summary for dependent variable

| N=1388 | b* | Std. Err. of b* | b | Std. Err. of b | t (1379) |
|---|---|---|---|---|---|
| Intercept | | | -0.026073 | 0.001608 | -16.2112 |
| Prod | -2.52837 | 0.026164 | -0.512563 | 0.005304 | -96.6366 |
| $HMG_1$ | 0.02116 | 0.008985 | 0.004168 | 0.001770 | 2.3546 |
| $HMG_2$ | 0.01693 | 0.009327 | 0.003346 | 0.001844 | 1.8150 |
| $HMG_3$ | -0.00080 | 0.008299 | -0.000157 | 0.001633 | -0.0960 |
| $HMG_4$ | 0.04287 | 0.011275 | 0.011948 | 0.003142 | 3.8023 |
| $E_p$ | 2.85701 | 0.043662 | 1.157669 | 0.017692 | 65.4347 |
| $E_q$ | -0.08315 | 0.040932 | -0.047577 | 0.023421 | -2.0314 |
| Cos Phi | 0.09614 | 0.020445 | -0.024913 | 0.005298 | -4.7023 |


From the Analysis of variance table (Table 5):
• Mean squares for regression is superior to the mean squares of the residual;
• The Fischer value is very high;
• P value is strictly inferior to 0.05.
The conclusion is that the Analysis of variance is validated.
**Table 5:** Analysis of Variance Table

| Effect | Analysis of Variance; DV: Ratio | | | | |
|---|---|---|---|---|---|
| | Sums of Squares | df | Mean Squares | F | p-value |
| Regress. | 49.31283 | 8 | 6.164104 | 1790.044 | 0.00 |
| Residual | 4.74865 | 1379 | 0.003444 | | |
| Total | 54.06149 | | | | |



Multiple linear regression has concluded that most of the experimental results are highly
adjusted and the model is explanatory, taking into consideration the value of $R^2$ and the
value standard error estimate which is very low (table 6).
**Table 6:** Summary statistics

| Statistic | Value |
|---|---|
| R | 0.95507172 |
| $R^2$ | 0.91216198 |
| Adjusted $R^2$ | 0.9116524 |
| Fischer (8.1379) | 1790 |
| P | 0 |
| Standard error of estimate | 0.0586817579 |

The final model is expressed by the equation below:
Rratio=2.85701 Ep – 2.52837Prod + 0.09614 Eq – 0.08315 CosPhi + 0.04287 HMG1 + 0.02116
HMG2                                                                                      Eq 2
**2.2. Technical Interpretations:**
From this analysis, this study concluded that in order to improve the ratio per produced
cubic meter, decision takers have to act firstly on reducing the active energy consumed by
the pumps. As the functioning point of the pumps is already set by the characteristics of the
system, we can only reduce the active energy by improving the efficiency of the pumps.
The ratio is negatively correlated with production means; i.e. there is an economy of
scale. It means the most the production increases the least the ratio is.
The operating hours of the pumps 1 and 4 are positively correlated, which means that
the more we use them the higher the ratio gets, so we'd better use the other groups,
especially the pump 3, and if there is an operation of renovation of the pumping station, it is
recommended to start with changing the pumps 1 and 4.
The model which is elaborated in this study has a standard error of estimate of 0.05 and
due to the lack of previous studies using multiple linear regression, we compared the results
with a study involving five data-mining approaches (Kusiak, 2013). The five data mining
approaches are the multi-layer, perceptron, neural network (MLP), the boosted-tree
(regression) algorithm (BT), the random-forest algorithm (RF), the support-vector machine
(SVM), and the k-nearest neighbour algorithm. These approaches had all provided more
than 90% of accuracy which is the case in the model of this study.
**CONCLUSION:**

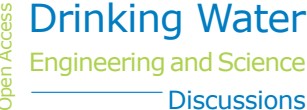

A Linear multiple regression was conducted to assess and study the influence of
multiple parameters on the ratio of the consumption of the energy per cubic meter water,
involved in a water pumping station.
This unique approach has allowed determining the real response of the system relying
on data that is measured over a 4 years period. Modelling the ratio will be a tool to take
decisions on which pump should the work be done first. This method combined with a cash
flow analysis, can help to take decisions on establishing priorities in case of renovations, to
change the pumps 1 and 4 with more efficient pumps.

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
