# Peer review of "Modeling of pump performance in a water pumping plant Fouad LAAJINE1,2, Mohammed MACHKOR2, Driss MAZOUZI1 1LRNE, multidisciplinary faculty of Taza, Sidi Mohamed Ben Abdellah University, Fez, Morocco 2National Office of Electricity and Drinking Water Morocco Correspondence: Driss MAZOUZI (driss.mazouzi@usmba.ac.ma)"

_Drinking Water Engineering and Science, 2021_

## Referee Comment (RC1)

**Review report "Modeling of pump performance in a water pumping plant" by Fouad Laajine, Mohammed Machkor and Diss Mazouzi**

**General comments**
The manuscript describes the modelling of a drinking water pumping station using multiple linear regression to model the kWh/m3 ratio depending on the input parameters. It finishes with the technical interpretation of the outcome of the model.
Although the approach is quite original, as it takes into account the real behaviour of the system, major revisions are required. There are many unclear sections in the manuscript, the Results and Discussion section is too concise, the English language has to be improved (the wording, many typo's, inconsistent use of capital letters (e.g. Multiple Linear regression, Multiple linear regression)) and the Tables and Figures are not well explained in the manuscript. The manuscripts looks like a short report and not as a scientific manuscript as any reference to other scientific papers, with the same approach or alternative approaches, is missing. This should be included in the introduction and discussion sections. In the present form it cannot be accepted.

**Specific comments**

*Abstract*
Line 15: what is meant by "real behaviour"?
Line 16: First should be first
Line 18: mention the input parameters
Line 19: what is meant by "phenomenon"?

Introduction
In the introduction any reference to other scientific papers dealing with optimizing pumping stations is missing.
Line 35: What is the current 2023 agenda?
Line 40: pumps account for 805 to 90% of the energy consumption, this depends on many factors (surface water or ground water, transport differences, flat or mountain region, etc).

*Materials and Methods*
- Lines 61-62: nice figure, but how does this research fits in this figure? To which category is it connected?
- line 86: tank RCMO? What does RCMO mean?
- line 87: probably the capacity of the pump is 457 m3/h?
- Lines 89-90, Figure 5: Are the ND of the suction line and the discharge line correct? They are not in line with the text I lines 79-82.
- Lines 97-103: the parameters should be defined in more detail. It is a list of parameters, while in the model eight input parameters are used: I assume the last (HMGI) covers four pumps? Be precise.
- Line 110: Y is the output variable.
- Lines 117-118: Table 1 is not clear. Just include the objective (not "1"), the variables (not "8", the responses (not "1").
- Line 119: Rephrase: Table 1 shows instead of The table above.
- Lines 122-123, Table 2: mean and standard deviation over the 4-years period?
- Line 123: Rephrase: Table 2 shows then mean and standard deviation

- Lines 125-126, Figure 6: What does this figure shows? What is on the Y-axis? What do the markers * and # mean?
- Line s128-129 Figure 7: This is a strange representation of a box plot. What is on the Y-axis? What do the p-values mean?

*Results and Discussion*
- there should be references to other studies (see also comment in the introduction). This is only a bullet-list of the main observations without any discussion. Please rewrite.
- Line 135: what is P?
- Line 138: What is Q?
- Lines 150-155: I suppose that "ratio" is kWh/m3?
- Line s156-157, Table 5: Table 5 is not clear, needs to be explained. Two situations, b and b*? not clear.
- Line 166: Multiple linear regression has *shown* that….
- line 167: Not clear, what is meant by "adjusted"?
- lines 174-191: Technical interpretation is nice, but it only deals with this case. Compariosn should be made with other approaches described in literature. Only one comparison is made in line 187 (five data-mining approaches), but it is not discussed whether this comparison is allowed.

*Conclusions*
- They should be rewritten. It is now just one sentence what the study was about and a couple of recommendations. What can be concluded from the research? Does the approach work? Is it different from other approaches? Etc.

**Principle criteria**

Scientific significance: fair
Scientific quality: poor
Presentation quality: poor

---

## Referee Comment (RC3)

Dear Dr. Grabowski,

Please find enclosed the revised version for the manuscript entitled "**Modeling of pump performance in a water pumping plant**" (Manuscript ID: ID 626534) as well as the answers to the referees questions.

We thank the reviewers for the valuable comments and their positive feedback. The revised version of the manuscript addresses the corrections suggested by the referees and clarifies the raised concerns.

Looking forward to hearing from you.

Sincerely,

Dr D. MAZOUZI

**Review report "Modeling of pump performance in a water pumping plant" by Fouad Laajine, Mohammed Machkor and Diss Mazouzi**

**General comments**

The manuscript describes the modelling of a drinking water pumping station using multiple linear regression to model the kWh/m3 ratio depending on the input parameters. It finishes with the technical interpretation of the outcome of the model.

Although the approach is quite original, as it takes into account the real behaviour of the system, major revisions are required. There are many unclear sections in the manuscript, the Results and Discussion section is too concise, the English language has to be improved (the wording, many typo's, inconsistent use of capital letters (e.g. Multiple Linear regression, Multiple linear regression)) and the Tables and Figures are not well explained in the manuscript. The manuscripts looks like a short report and not as a scientific manuscript as any reference to other scientific papers, with the same approach or alternative approaches, is missing. This should be included in the introduction and discussion sections. In the present form it cannot be accepted.

**Response:** Thanks for your kind reminders.

- We revised the title of our paper, new title: Multiple Linear Regression Analysis of Pumps Performance in Water Pumping Plants.

- We revised, introduction, all the sentences, new reference... of the paper who you find in attached a new paper with all correction asked.

- We hope that the manuscript has been improved towards after this revision.

**Specific comments**

*Abstract*

Line 15: what is meant by "real behaviour"?

**Response:** By using real-time-data, we revised the sentence as follow: "In this context, the main objective of this study was to model accurately the energy consumption of pumping systems in order to optimize the whole water supply system, thus improving its efficiency, especially in the case of a limited renovation"

Line 16: First should be first

**Response:** We revised the sentence by new version

Line 18: mention the input parameters

**Response:** The new sentence as follow: "For this purpose, Multiple Linear Regression was fitted to model the produced kWh/m3 ratio according to the following parameters, active and reactive energies, the daily produced water volume, the power factor (Cosj), and the operating time of each pump"

Line 19: what is meant by "phenomenon"?

**Response:** By the consumption, the new sentence as follow: "The final model describes accurately the consumption per cubic meter produced ($R^2$=0.91)"

Introduction

In the introduction any reference to other scientific papers dealing with optimizing pumping stations is missing.

**Response:** Thanks for your nice reminder. We provided the following citations to support this statement.

Adamowski, J., Fung Chan, H., Prasher, S. O., Ozga-Zielinski, B., Sliusarieva, A.: Comparison of multiple linear and nonlinear regression, autoregressive integrated moving average, artificial neural network, and wavelet artificial neural network methods for urban water demand forecasting in Montreal, Canada WATER RESOURCES RESEARCH, VOL. 48, W01528, doi:10.1029/2010WR009945, 2012.

Kusiak, A., Zeng, Y., Zhang, Z. : Modeling and analysis of pumps in a wastewater treatment plant : A data-mining approach, Engineering Applications of Artificial Intelligence., 26, 7, 1643-1651, 2013.

Shankar, A., Umashankar, V. K., Paramasivam, S., Norbert., S. H. : A comprehensive review on energy efficiency enhancement initiatives in centrifugal pumping system, Applied Energy, Elsevier, 181, C, 495-513, 2016

Carravetta, A. ; Giugni, M. ; Malavasi, S. : Application of Innovative Technologies for Active Control and Energy Efficiency in Water Supply Systems. Water, 12, 3278. https://doi.org/10.3390/w12113278, 2020.

Ostfeld, A. and Tubaltzev, A. : Ant Colony Optimization for Least Cost Design and Operation of Pumping and Operation of Pumping Water Distribution Systems. Journal of Water Resources Planning and Management, 134, 107-118. http://dx.doi.org/10.1061/(ASCE)0733-9496(2008)134:2(107), 2008.

Puleo, V., Morley, M., Freni, G., Savić, D., Multi-stage linear programming optimization for pump scheduling Procedia Engineering, 70, 1378-1385, 2014.

Plappally, A.K., Lienhard, J.H. V.: Energy requirements for water production, treatment, end use, reclamation, and disposal Renewable and Sustainable Energy Reviews 16, 4818–4848, 2012.

Rothausen, S., Conway, D. Greenhouse-gas emissions from energy use in the water sector. Nature Clim Change 1, 210–219, https://doi.org/10.1038/nclimate1147, 2011.

Wu, P., Lai, Z., Wu, D., Wang, L.: Optimization research of parallel pump system for improving energy efficiency, Journal of Water Resources Planning and Management, 141, 8, 2015.

Zhou, Y., Lee, E. W. M., Wong, L. T., & Mui, K. W. : Environmental evaluation of pump replacement period in water supply systems of buildings. Journal of Building Engineering, 40, 102750, 2021.

Line 35: What is the current 2023 agenda?

**Response:** The new sentence as follow: To reach the sixth Sustainable Development Goal (SDG6) that aims to generalize the access to drinking water supply, the water production cost mustn't impact its price which should stay affordable to the population

Line 40: pumps account for 805 to 90% of the energy consumption, this depends on many factors (surface water or ground water, transport differences, flat or mountain region, etc).

 **Response:** We provided an explanation in Subsection

"Pumping processes consume the largest fraction of total energy (Plappally and Lienhard, 2012). The pumps consumption often presents 80% to 90% of the total energy consumption (Sarbu, 2016). However, this consumption may depend on many factors such as surface water or ground water, transport differences, flat or mountain regions, etc (Rothausen and Conway, 2011) (Plappally and Lienhard, 2012)."

**Materials and Methods**

- Lines 61-62: nice figure, but how does this research fits in this figure? To which category is it connected?

**Response:** Thank you very much. We don't think so. Figure 2 is a figure of system energy efficiency

- line 86: tank RCMO? What does RCMO mean?

**Response:** Tank RMC0 Þ tank destined to provide the water treatment plant with raw water, it is called RMC0 and having a capacity of 1000 $m^3$

- Lines 89-90, Figure 5: Are the ND of the suction line and the discharge line correct? They are not in line with the text I lines 79-82.

**Response:** ND Þ nominal diameter

- Lines 97-103: the parameters should be defined in more detail. It is a list of parameters, while in the model eight input parameters are used: I assume the last (HMGI) covers four pumps? Be precise.

**Response:** HMG Þ  HMG1: the pump operating time "1",

HMG2: the pump operating time "2",

HMG3: the pump operating time "3",

HMG4: the pump operating time "4".

- Line 110: Y is the output variable.

**Response:** Thank you very much for the reminder, Y is the output variable

- Lines 117-118: Table 1 is not clear. Just include the objective (not "1"), the variables (not "8", the responses (not "1").

**Response:** Thank you very much for the reminder, we changed it as follows: "The effect of eight variables on the produced Kwh/m$^3$ ratio was evaluated (Table 1). Of note, 1388 experiments were conducted during 4 years."

- Line 119: Rephrase: Table 1 shows instead of The table above.

**Response:** Thank you very much for the reminder.

- Lines 122-123, Table 2: mean and standard deviation over the 4-years period?

**Response:** Thank you very much for the reminder. The mean and standard deviation during period 2015 and 2018

- Line 123: Rephrase: Table 2 shows then mean and standard deviation

**Response:** Thank you very much for the reminder. The mean and standard deviation during period 2015 and 2018

**Results and Discussion**

- there should be references to other studies (see also comment in the introduction). This is only a bullet-list of the main observations without any discussion. Please rewrite.

**Response:** Thank you very much your comments. We have read your comments carefully and tried our best to address them one by one, especially in results and discussion section. We hope that the manuscript has been improved towards after this revision.

- Line 135: what is P?

**Response:** P is the active energy consumed by the pumping station (measured by a wattmeter)

- Line 138: What is Q?

 **Response:** Q is the reactive energy consumed by the pumping station (measured by a wattmeter)

- Lines 150-155: I suppose that "ratio" is kWh/m3?

**Response:** yes, is electricity consumption by the pumps per $m^3$ of water consumption

- Line s156-157, Table 5: Table 5 is not clear, needs to be explained. Two situations, b and b*? not clear.

**Response:** Thank you for your nice reminder. We revised most of the table 4 and 5 captions to make them clearer. (b*) is Standardized regression coefficient and (b) is regression coefficient

- Line 166: Multiple linear regression has *shown* that….

**Response:** Thank you very much for pointing this out. We revised the sentence as follows:

"Besides that, R-square and Adjusted R-square statistic of this model was 0.91 were found and the value standard error estimate statistically significant at the level of 0.05, as shown in Table 6. In view of these results, we can say that the parameters studied are highly variable in the research area, thus, they also confirm the performance of the developed model."

- line 167: Not clear, what is meant by "adjusted"?

**Response:** is Adjusted R-square

- lines 174-191: Technical interpretation is nice, but it only deals with this case. Comparison should be made with other approaches described in literature. Only one comparison is made in line 187 (five data-mining approaches), but it is not discussed whether this comparison is allowed.

**Response:** Thank you for your comments. We have gone through your comments carefully and tried our best to address them one by one. We hope the technical interpretation section has been improved accordingly.

- Lines 125-126, Figure 6: What does this figure shows? What is on the Y-axis? What do the markers * and # mean?

- Line s128-129 Figure 7: This is a strange representation of a box plot. What is on the Y-axis? What do the p-values mean?

**Response:** Thank you for the nice reminder. We combined Figure 6 and 7 into one Figure 6 (A and B). The Y-axis is consumption of water ($10^3$ $m^3$). The markers * and #

was (*: consumption variation through the years; #: consumption variation through the months)

*Conclusions*

- They should be rewritten. It is now just one sentence what the study was about and a couple of recommendations. What can be concluded from the research? Does the approach work? Is it different from other approaches? Etc.

**Response:** Thank you very much for pointing this out. We revised the sentence of conclusions in new version manuscript:

**Principle criteria**

Scientific significance: fair

Scientific quality: poor

Presentation quality: poor

**Response:** Thank you very much for your previous comments that helped us improves this manuscript. The authors wish that the revised version of the manuscript addresses the corrections suggested by the referees and clarifies the raised concerns.

---

## Referee Comment (RC4)

REVIEW PAPER DWES-2021-16 (16 March 2022)

The paper illustrates a statistical analysis of operation of raw pumping station transporting the water from the source to the treatment plant, through a transmission main of approximately 3 km, based on the series of 4-year measurements of basic operational parameters.

GENERAL COMMENTS:

1. It is a case study-based paper without any specific scientific contribution.
2. I see no novelties claimed by the authors documented with any sufficient literature study.
3. The case could possibly be presented as a practitioner's paper but much is to be desired to bring it even to that level.
4. The background, the descriptions of the methodology, and the discussions and conclusions are pretty meagre. The whole structure of the paper is actually rather weak.
5. Although the text is not difficult to read, a further revision of English and explanations of used abbreviations is needed.
6. In this version, I cannot recommend the paper for publishing.

SPECIFIC COMMENTS (attached below the cut parts of the paper)

[Figure]

Drinking Water
Engineering and Science
Open Access                                    Discussions

**Modeling of pump performance in a water pumping plant**

Fouad LAAJINE[1,2], Mohammed MACHKOR[2], Driss MAZOUZI[1]

[1]LRNE, multidisciplinary faculty of Taza, Sidi Mohamed Ben Abdellah University, Fez, Morocco
[2]National Office of Electricity and Drinking Water Morocco

Correspondence: Driss MAZOUZI (driss.mazouzi@usmba.ac.ma)

SC 01: The title of the paper is not accurate description of the contents. I see no modelling component; it is a statistical analysis. Secondly, the term 'Water Pumping Plant' is confusing. I first thought that it was about clear water pumping station as an integral part of the water treatment plant, which is not the case. I would add the case study title to the revised paper title of pumping systems. The main objective of this study is to produce a model which reflects the real behaviour of a pumping system to help in taking decisions on which pump to use

First and which one to replace in case of a limited renovation. In order to do so, Multiple

SC02: I see no model in the study. It is a formula for statistical regression derived from the measured operational parameters. Not more, not less. I would certainly not understand how is that formula used for definition of replacement strategies. What is meant with 'limited renovation'? All this is not explained in the paper. English spelling: 'First' with capital 'F'?

Pumps account for 80% to 90% of the energy consumption (Sarbu, 2016). By achieving
energy efficiency improvements measures, we can reduce the consumption by at least 25%
(Moreira, 2013). Very few studies were conducted before to simulate the real behaviour of
pumping systems and evaluate the influence of parameters such as the aging of the
components, which can induce a reduction of the pumps performance for up to 12% (Kaya,
2008).

SC03: It is awkward to generalize any percentages referred from the literature because these normally emerge from some cases i.e. under specific conditions, which are not elaborated here. The pump ageing is interesting aspect, but it is not defined in the paper. How do we measure/monitor it? Was this included in the objective?

[Figure]

**Figure 1:** Energy and Hydraulic flows in a WSS

SC04: The drawing layout is confusing. It is mostly close to a water treatment plant. CWSS abbreviation does not stand because that one would also include transport and distribution infrastructure. On the other hand, the water and energy losses are indicated. Where they are originating from?

**1.4. Modelling:**
The aim of this study is to use Multiple linear regression, a wide popular technique to
predict an output from a range of inputs. MLP model with multiple input variables can be
expressed as following (Longo, 2016):

SC05: Why talking about the aim of the study in this place? What is the difference between the aim and objective? What is the exact meaning of MLP (should it be MLR?). English spelling: should be (' a widely popular technique'; 'Multiple linear regression' all words should start with capitals.

**Table 1:** Problem characteristics

| Objective of the study | The effects |
|---|---|
| Number of Variables | 8 |
| Number of experiments | 1388 |
| Number of the coefficients | 8 |
| Number of responses | 1 |

The table above summaries the objective of the study evaluating the effects of 8

variables on the response, which is the ratio of Kwh/m$^3$ produced. To get enough data, 1388

experiments were conducted during a period of 4 years.

SC06: The objective spelled in line 14 was to produce a model. Here it states that it is about 'the effects' (of what?). The table is confusing i.e. needs more elaboration:  the difference between the variables and coefficients, what is meant with number of responses, etc.

In the graphs below the distribution of the parameters is represented.

[Figure]

**Figure 6:** Representation of the variation of the consumptions through the years

SC07: Units are missing on Y-axis. Also, what is meant with 'Consumption'? Looking to the system layout in Fig. 4, it is more about a 'Production' in fact.

[Figure]

**Figure 7:** Box plots of the consumptions through the years

SC08: The same comment as SC07. Moreover, the meaning of P is not explained.

From the table of the regression summary (Table 4) it is conclude that the factors
influencing the ratio in a descending order are:

• Ratio is positively correlated with the active energy consumed by the pumps;

• Ratio is negatively correlated with the production;

• Ratio is positively correlated with the CosPhi;

• Ratio is negatively correlated with the reactive energy consumed by the
   pumps;

• Ratio is positively correlated with the operating hours of the pumps 1 and 4.

**Table 4:** Regression summary for dependent variable

| N=1388 | b* | Std. Err. of b* | b | Std. Err. of b | t (1379) |
|---|---|---|---|---|---|
| Intercept | | | -0.026073 | 0.001608 | -16.2112 |
| Prod | -2.52837 | 0.026164 | -0.512563 | 0.005304 | -96.6366 |
| $HMG_1$ | 0.02116 | 0.008985 | 0.004168 | 0.001770 | 2.3546 |
| $HMG_2$ | 0.01693 | 0.009327 | 0.003346 | 0.001844 | 1.8150 |
| $HMG_3$ | -0.00080 | 0.008299 | -0.000157 | 0.001633 | -0.0960 |
| $HMG_4$ | 0.04287 | 0.011275 | 0.011948 | 0.003142 | 3.8023 |
| $E_p$ | 2.85701 | 0.043662 | 1.157669 | 0.017692 | 65.4347 |
| $E_q$ | -0.08315 | 0.040932 | -0.047577 | 0.023421 | -2.0314 |
| Cos Phi | 0.09614 | 0.020445 | -0.024913 | 0.005298 | -4.7023 |

SC09: There is a repetitive mentioning of a 'ratio' but no explanation which one.

SC10: To which extent is the statistical analysis giving surprising or logical correlations? Could the relations be known even without doing it? The bullets only read the table, without real discussions.

> 181  The operating hours of the pumps 1 and 4 are positively correlated, which means that
> 182  the more we use them the higher the ratio gets, so we'd better use the other groups,
> 183  especially the pump 3, and if there is an operation of renovation of the pumping station, it is
> 184  recommended to start with changing the pumps 1 and 4.

SC11: The pumping station has four identical units. So obviously, shuffling their operation schedules does not interfere with the target hydraulic performance while it is 'healthy' for the lifetime of each pump. This is a common engineering logic. I do not understand what more we learn from the results in the tables in order to operate the pumps differently? The interpretation of the results is very superficial.

> 185  The model which is elaborated in this study has a standard error of estimate of 0.05 and
> 186  due to the lack of previous studies using multiple linear regression, we compared the results
> 187  with a study involving five data-mining approaches (Kusiak, 2013). The five data mining
> 188  approaches are the multi-layer, perceptron, neural network (MLP), the boosted-tree
> 189  (regression) algorithm (BT), the random-forest algorithm (RF), the support-vector machine
> 190  (SVM), and the k-nearest neighbour algorithm. These approaches had all provided more
> 191  than 90% of accuracy which is the case in the model of this study.

SC12: I see no evidence of any comparison in the paper. How can I trust?

> 196  This unique approach has allowed determining the real response of the system relying
> 197  on data that is measured over a 4 years period. Modelling the ratio will be a tool to take
> 198  decisions on which pump should the work be done first. This method combined with a cash
> 199  flow analysis, can help to take decisions on establishing priorities in case of renovations, to
> 200  change the pumps 1 and 4 with more efficient pumps.

SC13: What is 'unique'? What is meant with 'real response'? How do we really benefit from the measurements done to improve the operation of the pumps?

SC14: The suggested financial considerations should already be added to improve the substance of the paper.

SC15: I do still do not understand the rationale to replace 'the pumps 1 and 4 with more efficient pumps'. Why they are currently worse than pumps 2 and 3 when they are all identical. Again, too superficial discussion of the results.

---

## Author Comment (AC4)

REVIEW PAPER DWES-2021-16 (16 March 2022)

The paper illustrates a statistical analysis of operation of raw pumping station transporting the water from the source to the treatment plant, through a transmission main of approximately 3 km, based on the series of 4-year measurements of basic operational parameters.

Thank you for carefully reviewing our manuscript and providing a referee comment in the public review process. Your constructive comments will improve our manuscript. Please find our answers below in red

GENERAL COMMENTS:

1. It is a case study-based paper without any specific scientific contribution.

Response: Our specific scientific contribution in the paper is that, basically was fitted to model the produced energy consumption for pump efficiency of the pumping station and predict the next renovation of the pumps involved in a water pumping station. We believe the section in the aim of the introduction can be strengthened by highlighting these points raised by the reviewer:

"In this context, this work introduces a study based on statistical modelling by using multiple regression method to analyze the key factors affecting the efficiency of the pumping for drinking water production. Hence, this paper presents the results targets the pumping system of the Bab Louta drinking water production located in the province of Taza, Morocco. In this perspective, a Multiple Linear Regression (MLR) was fitted to model the produced energy consumption kWh/m$^3$ ratio according to the input key-parameters by using Real-Time-Data based on the collected data such as: the active energy and reactive energy consumed by the pumping station, the daily produced volume, the power factor and the pump operating time. Of other objectives, this study can also predict the next renovation of the pumps involved in a water pumping station."

2. I see no novelties claimed by the authors documented with any sufficient literature study.

Response: We will make the necessary revisions.

3. The case could possibly be presented as a practitioner's paper but much is to be desired to bring it even to that level.

4. The background, the descriptions of the methodology, and the discussions and conclusions are pretty meagre. The whole structure of the paper is actually rather weak.

Response: Thanks for your kind reminders. We revised, introduction, all the sentences, new reference... of the paper who you find in attached a new paper with all correction asked. We hope that the manuscript has been improved towards after this revision.

5. Although the text is not difficult to read, a further revision of English and explanations of used abbreviations is needed.

Response: We agree with the reviewer. We believe a reorganization and enhancement for next new paper after revision.

6. In this version, I cannot recommend the paper for publishing.

Response: These comments and suggestions will undoubtedly improve the impact and utility of our paper

SPECIFIC COMMENTS (attached below the cut parts of the paper)

SC 01: The title of the paper is not accurate description of the contents. I see no modelling component; it is a statistical analysis. Secondly, the term 'Water Pumping Plant' is confusing. I first thought that it was about clear water pumping station as an integral part of the water treatment plant, which is not the case. I would add the case study title to the revised paper title

Response: Thank you very much for your comment. We have changed the title to "Statistical Modelling Based on Multiple Linear Regression Analysis Method of Pumps Performance in the Pumping for Drinking Water Production"

SC02: I see no model in the study. It is a formula for statistical regression derived from the measured operational parameters. Not more, not less. I would certainly not understand how is that formula used for definition of replacement strategies. What is meant with 'limited renovation'? All this is not explained in the paper. English spelling: 'First' with capital 'F'?

Response: Thanks for your kind reminders. We revised the sentence as follows:

"Regarding water utilities, most of their operating costs are related to energy consumption, especially pumping systems consumption. In this context, the main objective of this study was to model accurately by using data statistical analysis the energy consumption of pumping systems in order to optimize the whole water supply system, thus improving its efficiency, especially in the case of a limited renovation".

SC03: It is awkward to generalize any percentages referred from the literature because these normally emerge from some cases i.e. under specific conditions, which are not elaborated here. The pump ageing is interesting aspect, but it is not defined in the paper. How do we measure/monitor it? Was this included in the objective?

Response: Thanks for your kind reminders. We agree with the reviewer's assessment. Accordingly, we revised the sentence as follows:

"Pumping processes consume the largest fraction of total energy (Plappally and Lienhard, 2012). The pumps consumption often presents 80% to 90% of the total energy consumption (Sarbu, 2016). However, this consumption may depend on many factors such as surface water or ground water, transport differences, flat or mountain regions, the pump ageing, etc (Rothausen and Conway, 2011) (Plappally and Lienhard, 2012). Thus, by achieving energy efficiency improvements measures, we may reduce this consumption by 25% (Moreira, et al., 2013). In this context, few studies were interested in modeling pumping systems and evaluated the influence of parameters such as the aging of the components, which can reduce the performance of the pump by up to 12% (Durmus, et al., 2008)."

SC04: The drawing layout is confusing. It is mostly close to a water treatment plant. CWSS abbreviation does not stand because that one would also include transport and distribution infrastructure. On the other hand, the water and energy losses are indicated. Where they are originating from?

Response: We agree with the reviewer. We believe a reorganization and enhancement of this section can improve the text:

"Conventional water supply systems (WSS) consist of sets of structures and facilities providing products with a suitable quantity and quality for domestic and industrial use (Luna et al., 2019). Basically, the energy consumption in WSS is closely connected with water demand, generally, this consumption is associated to pumping systems and represents the largest share of energy consumption in the entire water sector. Therefore, it is interesting to develop energy and hydraulic model for to achieve high energy efficiency. To do this, the WSS systems should be evaluated a day-to-day analysis in terms of mass and energy (Figure 1)."

SC05: Why talking about the aim of the study in this place? What is the difference between the aim and objective? What is the exact meaning of MLP (should it be MLR?). English spelling: should be (' a widely popular technique'; 'Multiple linear regression' all words should start with capitals.

Response: Thank you very much for the reminder. We have made revisions accordingly.

"Regression analysis, is a statistical technique that uses several explanatory variables to predict the outcome of a response variable. This study uses a Multiple Linear Regression (MLR) analysis to predict an output from a range of inputs. This means that the MLR analysis would make it possible to obtain a relationship between the key-parameters associated to the pumping for drinking water production and the produced $kWh/m^3$ ratio costs. MLR model with multiple input variables can be expressed as follows (Longo et al., 2016):"

SC06: The objective spelled in line 14 was to produce a model. Here it states that it is about 'the effects' (of what?). The table is confusing i.e. needs more elaboration: the difference between the variables and coefficients, what is meant with number of responses, etc.

Response: Thank you very much for your previous comments that helped us improve this manuscript. We revised the sentence as follows:

"In order to assess the influence of the included parameters on the cubic meter ratio produced by the pumping station, the following key-parameters (input variables) were considered: the active energy ($E_P$), the reactive energy ($E_Q$), the daily produced volume (V), the power factor ($Cos\varphi$), and the operating time of each pump ($HMG_i$). Therefore, the principal parameter analysis is used to establish the evaluation model to achieve more objective and accurate analysis.

The effect of eight variables on the produced Kwh/m$^3$ ratio was evaluated. Of note, 1388 experiments were conducted during 4 years. The set of analysis data is summarized in Table 1."

**Table 1:** Problem characteristics

| Objective of the study | The effect of the variables on the ratio KWh/m$^3$ |
|---|---|
| Number of Variables | 8 |
| Number of experiments | 1388 |
| Number of the coefficients | 8 |
| Number of responses | 1 |

SC07: Units are missing on Y-axis. Also, what is meant with 'Consumption'? Looking to the system layout in Fig. 4, it is more about a 'Production' in fact.

SC08: The same comment as SC07. Moreover, the meaning of P is not explained.

Response: Thank you for the nice reminder. We combined Figure 6 and 7 into one figure (Figure 6). We revised the sentence as follows

[Figure]

**Figure 6:** Representation the dataset collected of the variation of the consumptions of water versus the months during period 2015 and 2018, (A) consumption variation through the months (*: P<0.05, consumption variation through the years; #: P<0.05, consumption variation through the months), (B) consumption variation through the years (P-value statistical analysis; Each point represents the consumption average of a given month during the year).

The dataset collected of the water production across the year for each year from 2015 to 2018 presented in figure 6. The knowledge of such curves allows the consumptions of water to be assessed. In this way, the day, mouth and annual yield of the energy consumption related to pump station can be estimated. For analysis of this data, it was noticed that there is a higher production which reflects a higher consumption of water during the summer months in province of Taza (Figure 6-A). On the other hand, Figure 6-B shows the evolution of the production through the years 2015 to 2018 and it has also the same increasing trend due to the continuous commissioning of new networks leading to a growing number of consumers. The p-value of these statistical analysis is approximately null, showing

that the explanation of the selected independent variables is statistically significant, considering a level of significance of 5%.

It can be seen from Table 1, 2 and Figure 6 that the original data has large differences and many influencing factors, and it is difficult to conduct comprehensive and systematic analysis by conventional methods. Therefore, the MLR analysis method is used to establish the evaluation model to achieve more objective and accurate analysis.

SC09: There is a repetitive mentioning of a 'ratio' but no explanation which one.

SC10: To which extent is the statistical analysis giving surprising or logical correlations? Could the relations be known even without doing it? The bullets only read the table, without real discussions.

Response: Thanks for your question, most of the update in this paper of this section:

"The correlation coefficients shown in a matrix (Table 3) are the results of statistical analyses for possible relationships between different parameters monitored. It was found that the studied variables are strongly correlated. Taking the above into the account in the energy consumption system of water supply, it is justified to include that:

- The active energy consumed by the pumping station was dependent on the production, reactive energy, pumps operating hours (4;1), power factor ($Cos\varphi$), $HMG_2$, and $HMG_3$ respectively.

- The reactive energy consumed is highly was dependents on the production, active energy, $Cos\varphi$, pumps operating hours (4;1;2;3) respectively.

- $Cos\varphi$ was dependent on the reactive energy, the production, active energy, pumps operating hours (4;1;3;2) respectively.

- Production was dependent on the active energy, the reactive energy pumps operating hours (4;1;2;3)."

SC11: The pumping station has four identical units. So obviously, shuffling their operation schedules does not interfere with the target hydraulic performance while it is 'healthy' for the lifetime of each pump. This is a common engineering logic. I do not understand what more we learn from the results in the tables in order to operate the pumps differently? The interpretation of the results is very superficial.

Response: Thanks for the comments. More revisions have been made to the relevant parts in the:

"The ratio is negatively correlated with production means; i.e., there is an economy of scale. It means the most the production increases the least the ratio is. It allows comparing the results using Multiple Linear Regression models with pumping station to avoid excessive energy consumption. The operating hours of the pumps 1 and 4 are positively correlated, which means that more these pumps are used higher more the ratio is higher. Therefore, it is recommended to use the pump 3, and if there is an operation of renovation of the pumping station, it is suggested to start with upgrading the pumps 1 and 4, which may also depend on the aging of these pumps. In the event of a new investment, the company can increase the capacity of the RMC0 storage tank which, according to the model, will decrease the significantly the ration and also allows a load shifting to the off-peak hours."

SC12: I see no evidence of any comparison in the paper. How can I trust?

Response: Thanks for your question, the new section quoted above by the authors in the paper show the comparison the results

"The model which was elaborated in this study, was successfully validated in the statistical analysis. It shows that the R-square statistic reaching 0.91, and a standard error of estimate of 0.05. Thus, due to the lack of previous studies using Multiple Linear Regression, we compared the results with a previous study involving five data-mining approaches (Kusiak et al., 2013). This study had for objective to model the energy consumption in a comparable case of a wastewater pumping station that has 3 pumps that transfer the wastewater to a treatment plant. Although there are differences between the flow capacities and the pressure with drinking water supply facilities but the approach remains the same. The five data mining approaches are the multi-layer, perceptron, neural network (MLP), the boosted-tree (regression) algorithm (BT), the random-forest algorithm (RF), the support-vector machine (SVM), and the k-nearest neighbor algorithm. These approaches had all provided more than 90% of accuracy which is the case in the model of this study. The benefit of our method goes beyond the control methods used in most of the optimization approaches which only provide a method to operate the system in an efficient way but don't account for other factors such as the aging of the pumps, factors that are crucial when upgrading the system.

This study had for objective to model the energy consumption in a comparable case of a wastewater pumping station that has 3 pumps that transfer the wastewater to a treatment plant. Although there are differences between the flow capacities and the pressure with drinking water supply facilities but the approach remains the same. The five data mining approaches are the multi-layer, perceptron, neural network (MLP), the boosted-tree (regression) algorithm (BT), the random-forest algorithm (RF), the support-vector machine (SVM), and the k-nearest neighbor algorithm. These approaches had all provided more than 90% of accuracy which is the case in the model of this study. The benefit of our method goes beyond the control methods used in most of the optimization approaches which only provide a method to operate the system in an efficient way but don't account for other factors such as the aging of the pumps, factors that are crucial when upgrading the system."

SC13: What is 'unique'? What is meant with 'real response'? How do we really benefit from the measurements done to improve the operation of the pumps?

SC14: The suggested financial considerations should already be added to improve the substance of

the paper.

Response: Thank you. It's a good question. The description is now revised to give better understanding:

"A Linear Multiple Regression was conducted to assess and study the influence of multiple parameters on the energy consumption ratio per cubic per cubic meter water, involved in a water pumping station.

   This unique approach has allowed evaluating the real response of the system relying on data that is measured over a 4 years period. Modelling the ratio will be a tool to take decisions on which pump should the work be done first. This method combined with a cash flow analysis, can help to take decisions on establishing priorities in case of renovations, to change the pumps 1 and 4 with more efficient pumps. To validate this model, we performed the performance test by determining the correlation R to show the link between the produced kWh/m$^3$ ratio and the following parameters such as active and reactive energies, the daily produced water volume, the power factor (Cosφ), and the operating time of each pump. the regression coefficients, thus validating the models. The final model describes accurately the consumption per cubic meter produced with a R-square statistic reaching 0.91.

   After this study, we retain that the developed model can predict the energy consumption ratio per cubic meter water, involved in a water pumping station. Thus, the model would be useful when the next renovation will be undertaken by the office which will conduct a replacement of the pumps in the year 2024, can more accurately and reasonably evaluate the efficiency pumping, according to the pumping unit model, motor power…

   Besides that, the above findings demonstrate the potential of method for solving real-time pump scheduling problems in large water distribution systems with many pumps. However, this requires further work with other metaheuristic methods such as Genetic Algorithms before relevant conclusion can be made".

SC15: I do still do not understand the rationale to replace 'the pumps 1 and 4 with more efficient pumps'. Why they are currently worse than pumps 2 and 3 when they are all identical. Again, too superficial discussion of the results

Response: We are grateful for this comment as it points to an important rationale of this study,

which concern the prediction to change the pumps. Typically, all the pumps are energy consumers. This is the reason why our final mathematical model includes the energy consumption of all pumps. However, it is of note that the energy consumption of these different pumps is not equivalent, particularly for pumps 1 and 4. For this reason, we have particularly underlined their energy consumption in the first version of this manuscript. In order to clarify this confusion in the revised manuscript, the discussion focusing only on 1 and 4 pumps was replaced by a discussion integrating the entire pumps.

The following paragraphs have been added to reflect this improved discussion:

"The ratio is negatively correlated with production means; i.e., there is an economy of scale. It means the most the production increases the least the ratio is. It allows comparing the results using Multiple Linear Regression models with pumping station to avoid excessive energy consumption, which is able to estimate the performance and to make a proper decision on the pumps. The results indicate the operating hours of the pumps 1 and 4 are positively correlated, which means that more these pumps are used higher more the ratio is higher. Therefore, it is recommended to use the pump 3, and if there is an operation of renovation of the pumping station, it is suggested to start with upgrading the pumps 1 and 4, which may also depend on the aging of these pumps. In the event of a new investment, the company can increase the capacity of the RMC0 storage tank which, according to the model, will decrease the significantly the ration and also allows a load shifting to the off-peak hours."